# Microband-Induced Plasticity in a Nb Content Fe–28Mn–10Al–C Low Density Steel

**Tao Ma, Jianxin Gao, Huirong Li, Changqing Li, Haichao Zhang and Yungang Li \***

College of Metallurgy &Energy, North China University of Science and Technology, Tangshan 063210, China;
matao2011@sina.com (T.M.); gjx@ncst.edu.cn (J.G.); lihuirong@ncst.edu.cn (H.L.);
lichangqing@stu.ncst.edu.cn (C.L.); zhanghaichao@stu.ncst.edu.cn (H.Z.)
\* Correspondence: liyungang59322@163.com or lyg@ncst.edu.cn

**Abstract:** Novel Fe–28Mn–10Al–C–0.5Nb low-density steel was developed and the room temperature tensile behavior in the solid solution state and the microstructure evolution process during plastic deformation were studied, aiming to clarify the dominant deformation mechanisms. The results show that the developed steel was fully austenitic with a low density of 6.63 g/cm$^3$ and fairly high stacking fault energy of 84 MJ/m$^2$. The present fully austenitic Fe–28Mn–10Al–C–0.5Nb low-density steel exhibited an excellent ultimate tensile strength of 1084 MPa and elongation of 37.5%; in addition, the steel exhibited an excellent combination of strength and ductility (i.e., the product of strength and ductility (PSE) could reach as high as 40.65 GPa%). In spite of the high stacking fault energy, deformed microstructures exhibited planar glide characteristics, seemingly due to the glide plane softening effect. The excellent combination of strength and ductility is attributed to plasticity induced by microbands and leads to the continuous strain hardening during deformation at room temperature. Moreover, the addition of Nb does not change the deformation mechanism and strengthening mechanism of Fe–Mn–Al–C low-density steel, and can optimize the mechanical properties of the steel.

**Keywords:** Fe–Mn–Al–C; Nb; deformation mechanisms; planar glide; microbands

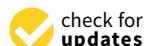

## 1. Introduction

The development of high-strength steel plates with high ductility and toughness for automotive manufacturing has been pursued for a long time. Nowadays, several advanced steel materials with a combination of strength and ductility (i.e., tensile strength (TS) × elongation) of 15,000–20,000 MPa% have been applied to automobile sheet steels such as gapless steel, high strength low alloys (HSLA), transformation induced plasticity steel (TRIP), twin-induced plasticity steel (TWIP), etc. [1,2]. However, due to rising fuel costs and restrictions on exhaust emissions, automakers have begun to devote attention to developing vehicles with low fuel consumption and high safety. As a solution, the applications of lightweight and safe steel to reduce the weight of cars is considered an efficient approach. However, the weight savings of commercially available automotive sheet steels are not effective due to the low alloying of light elements such as Al. Several recent investigations have revealed that Fe–Mn–Al–C steel, prepared by adding Al to high Mn austenitic steel, can be used in automobile manufacturing to realize automobile lightweight [3,4] due to its low density, excellent mechanical properties, formability, and weldability. Therefore, the development of Fe–Mn–Al–C low-density steels is widely of interest worldwide [5–7].

Compared with TRIP/TWIP steel, Fe–Mn–Al–C steel has lower density and better mechanical properties with considerable Al addition (e.g., 10%) [8–12]. These effects mainly result from the stacking fault energy (SFE) increase with the addition of Al in the high Mn austenitic steel and making the dislocation plane slip the main deformation mechanism. In addition, the addition of Al also induces dynamic precipitation of κ-carbide. The L′$_{12}$ type κ-carbides with FCC structure will precipitate by spinodal decomposition

of austenite. Through precipitation strengthening of κ-carbides, dislocation plane slip is prevented, and the mechanical properties of low density steel will be improved. For example, Sutou et al. [13] reported that the fine κ-carbides produced during the quenching process will make Fe–20Mn–(10,11)Al–(1.0,1.5)C low-density steel obtain high yield strength and tensile strength while maintaining high ductility. Kalashnikov [14] reported that when the Al content exceeds 7%, the austenitic Fe–Mn–Al–C steel after aging treatment is significantly strengthened due to the precipitation of nano-scale κ-carbides, with yield strength and tensile strength elevated. However, as reported by Sutou [13], Lu [15], and Liu [16], etc., due to the stress concentration at the grain boundary, coarse κ-carbides distributed along grain boundaries would reduce the ductility of low density steel [17,18]. Therefore, the size and distribution of κ-carbides need to be controlled to improve the mechanical properties of low-density steel. This limits the development and preparation of higher-performance austenitic Fe–Mn–Al–C low-density steel for automobiles, and also greatly increases its production difficulty and cost.

In order to better improve the tensile property, the present work aimed to develop a Nb alloyed Fe–Mn–Al–C low-density steel. First of all, Nb has a strong affinity with C, which consumes C atoms in low-density steel to form a very stable NbC phase [19], and the precipitation of coarse κ-carbides can be suppressed. Moreover, NbC precipitates uniformly distributed in austenite can be obtained, and the size of the austenite grain can be refined [20]. The mechanical properties of low density steel can be optimized by precipitation strengthening and fine grain strengthening through the precipitation of NbC. In addition, due to the strong stability of NbC, it can maintain the nano-scale particle size during the high-temperature aging process, and can effectively avoid the loss of ductility caused by the coarsening of the precipitated phase [21]. At present, there are less reports on the preparation of Fe–Mn–Al–C–Nb low density steel by Nb alloying, and the evolution of the microstructure of Fe–Mn–Al–C–Nb low density steel during plastic deformation has not attracted attention, and the strengthening mechanism is still unclear. In the present investigation, the correlation between room temperature tensile behavior and the deformed microstructure of Fe–Mn–Al–C–Nb low density steel was analyzed, in order to determine the plastic deformation mechanism and strengthening mechanism of this new product. These results may provide theoretical support for subsequent research and development.

## 2. Experimental Procedure

A 25 kg ingot was prepared by induction melting in an argon atmosphere. In order to obtain a plate-like shape, the ingot was forged and hot rolled, and a high temperature heat treatment was performed before and after machining to remove internal stress. The ingot was heated at 1150 °C for 2 h, and then forged into a slab with sectional dimensions of 80 mm × 40 mm. After homogenization at 1200 °C for 2 h, the slab was hot rolled to the plate of 5 mm thickness. After that, the hot rolled plate was solution treated at 950 °C for 1 h and water quenched to room temperature. This process is shown in Figure 1. The chemical composition of steel with error bars is shown in Table 1 and Figure 2. According to the Archimedes principle, the density of the Fe–Mn–Al–C–Nb low density steel, measured by densitometry (Byes-300A, Byes, Shanghai, China) is 6.63 g/cm$^3$, which is 15% lower than pure iron.

Tensile specimens with a section of 25 mm × 5 mm × 6 mm were machined from the hot-rolled plate, along the parallel to the rolling direction. The shape and size of the tensile specimens is shown in Figure 3, according to the GB/T 228.1-2010 [22] sub-size standard. The room temperature tensile test was carried out up to failure on the universal testing machine (INSTRON 3382, Instron, Norwood, MA, USA) at an initial strain rate of $1 \times 10^{-3}s^{-1}$. The microstructural evolution and microstructural evolution at the different strain levels were observed by interrupted tensile tests at predetermined strains. The microstructure examination and the analysis of the characteristics of the precipitated phase were performed on the optical microscope (OM, Leica DMi8, Leica, Wentzler, Germany), scanning electron microscope (SEM, FEI Quanta 650, Thermo Fisher

Scientific, Hillsboro, OR, USA), and transmission electron microscope (TEM, Tecnai G2 F20 S-TWIN, Thermo Fisher Scientific, Hillsboro, OR, USA). The specimens used for OM observation were mechanically polished and then etched with 4% Nital. The average grain size was determined by a Nano Measurer. Thin foils for TEM analysis were electro-polished by a twin-jet polishing using a mixture of 10% perchloric acid and 90% ethanol with an applied voltage of 20 V at −35 °C. TEM observations were performed at 200 kV. Phase constituents of the specimen were examined by an x-ray diffractometer (XRD, Rigaku, D/Max/PC, Rigaku Corporation, Tokyo, Japan) with Cu-Kα radiation, and scanning speed of 5°/min in the angle range of 40–100°.

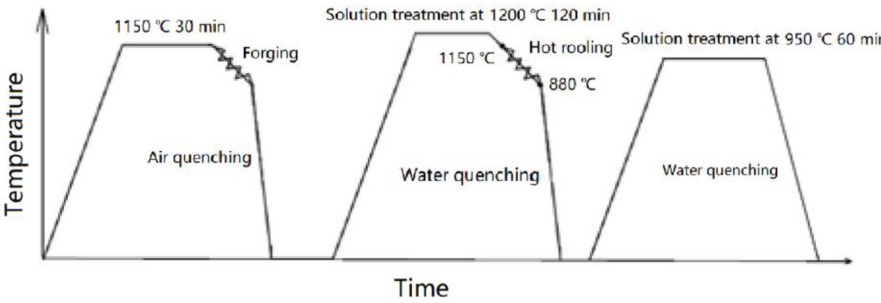

**Figure 1.** Test setup and the details of tensile specimens of the dog bone geometry.

**Table 1.** Chemical compositions of the designed steels.

| Composition | Al | Mn | C | Nb | S | P | Fe |
|---|---|---|---|---|---|---|---|
| Wt.% | 10.83 | 29.33 | 1.02 | 0.52 | 0.0028 | 0.0031 | Bal |

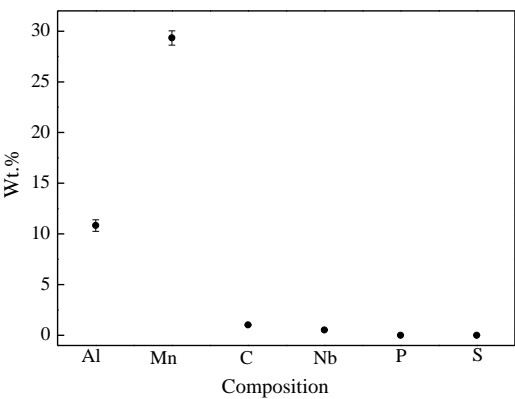

**Figure 2.** Chemical compositions of the designed steels with error bars.

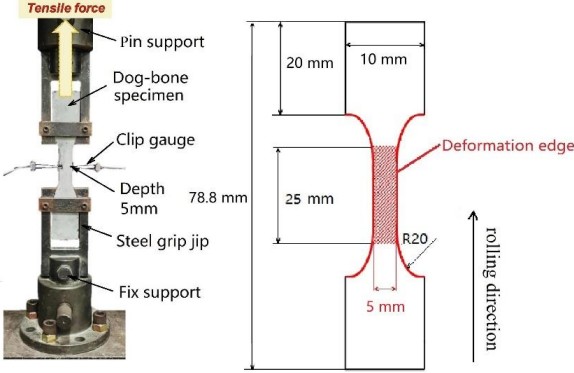

**Figure 3.** Test setup and the details of the tensile specimens of the dog bone geometry.

## 3. Results

### 3.1. Microstructure and Precipitate Morphology

The microstructure of the steel after solution treatment at 950 °C, followed by quenching, is shown in Figure 4a. Specimens had a stable austenite structure and very few annealing twins, and the grain size on average was 13.67 μm. The result of the XRD analysis is shown in Figure 4b, indicating the presence of face-centered cubic (fcc) peaks and that the sample was fully austenitic.

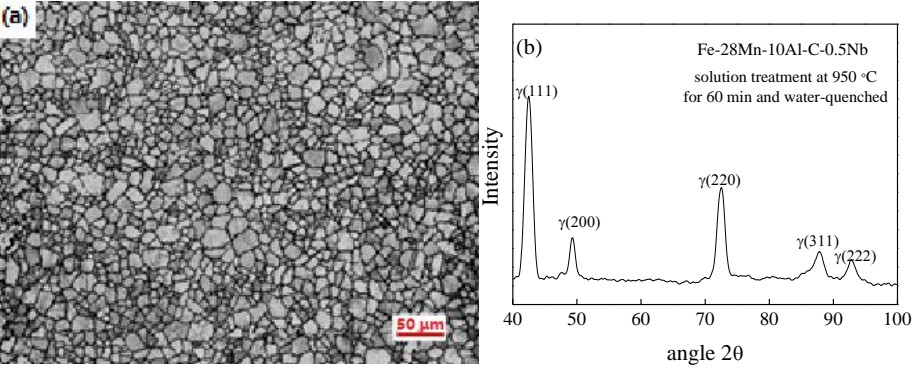

**Figure 4.** (**a**) Optical micrographs and (**b**) x-ray diffraction (XRD) profile of Fe–28Mn–10Al–C–0.5Nb steel after solution treatment at 950 °C followed by water-quenching.

Figure 5 shows the SEM morphologies and Energy Dispersive Spectrometer (EDS) analysis of the present steel. As can be seen in Figure 5, in the wake of adding Nb to low-density steel, precipitated phases formed, and were mainly distributed continuously along the austenite grain boundaries. The EDS element mapping shows that the distribution positions of Nb and C were almost the same, indicating that the precipitates were NbC. In order to further analyze the morphologies of the precipitated phases, Figure 6 shows the bright-field TEM morphologies of the steel. As shown by the bright-field TEM morphologies, the second phase precipitated in the steel after solution treatment, which had an ellipsoid shape. The precipitates were distributed in austenite grains and grain boundaries, with a particle size of 450 nm, and the EDS results showed that the precipitates were NbC.

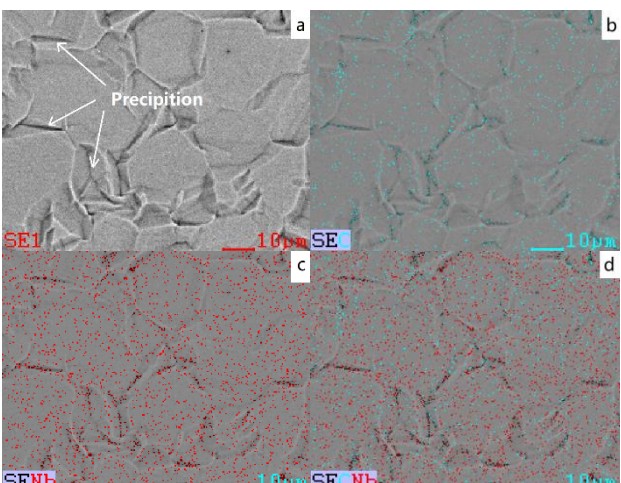

**Figure 5.** Scanning electron microscopy (SEM) image (**a**) and EDS element mapping (**b,c,d**) of Fe-28Mn–10Al–C–0.5Nb: (**b**) C, (**c**) Nb, and (**d**) C and Nb.

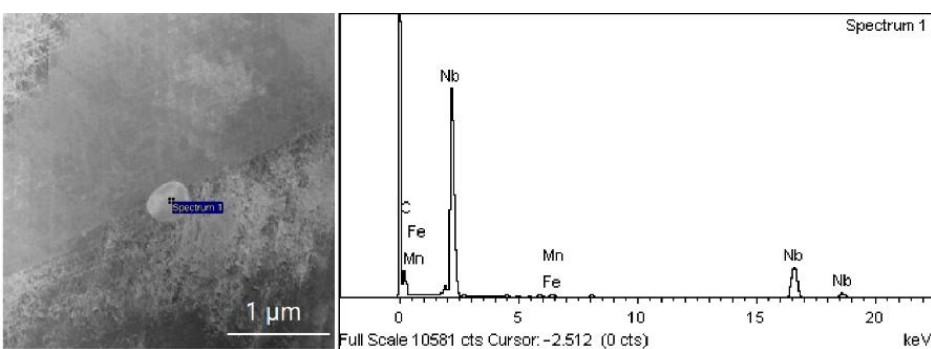

**Figure 6.** Transmission electron microscopy (TEM) bright-field morphologies and EDS analysis of Fe–28Mn–10Al–C–0.5Nb steel.

*3.2. Stacking Fault Energy*

Al has a great influence on the stacking fault energy (SFE) of Fe–Mn–Al–C low density steel. SFE determines the deformation mechanism of low-density steel [23,24], so will therefore affect the mechanical properties. Based on the thermodynamic model established by Olson–Cohen [25], the SFE ($\tau$) can be estimated as:

$$\tau = 2\rho\Delta G^{\gamma \rightarrow \varepsilon} + 2\sigma \tag{1}$$

$$\rho = \left(\frac{4}{3}\right)\left(\frac{1}{\alpha^2 N}\right) \tag{2}$$

where $\Delta G^{\gamma \rightarrow \varepsilon}$ is the Gibbs free energy for phase transition of $\gamma_{fcc}$; $\rho$ is the molar planar density of {111} plane; $\sigma$ is the interfacial energy of $\gamma/\varepsilon$; $\alpha$ is the lattice constant of $\gamma_{fcc}$; and N is Avogadro's number.

In a single-phase ($\varphi$) system composed of elements i and j, its Gibbs free energy $G^{\varphi}$ can be generally expressed as:

$$G^{\phi} = X_i G_i^{\phi} + X_j G_j^{\phi} + RT\left(X_i lnX_i + X_j \ln\right) + X_i X_j \Omega_{ij}^{\phi} \tag{3}$$

$$\Omega_{ij}^{\phi} = L_0^{\phi} + L_1^{\phi}\left(X_i - X_j\right) \tag{4}$$

$$G_{mg}^{\phi} = f\left(\frac{T}{T_{neel}}\right)RTln(\beta + 1) \tag{5}$$

where X is the mole fraction; $G^{\varphi}$ is Gibbs free energy of; $L_0$ is a temperature-based parameter; $L_1$ is a constant; $T_{neel}$ is Neel temperature; and $\beta$ is the magnetic moment. From Equations (3)–(5), The $\Delta G^{\gamma \rightarrow \varepsilon}$ of the $\gamma/\varepsilon$ transformation of austenitic steel composed of two components can be expressed as:

$$\Delta G^{\gamma \rightarrow \varepsilon} = X_i \Delta G_i^{\gamma \rightarrow \epsilon} + X_j \Delta G_j^{\gamma \rightarrow \epsilon} + X_i X_j \Omega_{ij}^{\gamma \rightarrow \epsilon} + \Delta G_{mg}^{\gamma \rightarrow \epsilon} \tag{6}$$

According to Equation (6) and regular solid solution model, the G of Fe–Mn–Al–C and Fe–Mn–Al–C–Nb multi-component low-density steel can be expressed as:

$$
\begin{aligned}
\Delta G^{\gamma \rightarrow \varepsilon} &= X_{Fe}\Delta G_{Fe}^{\gamma \rightarrow \epsilon} + X_{Mn}\Delta G_{Mn}^{\gamma \rightarrow \epsilon} + X_{Al}\Delta G_{Al}^{\gamma \rightarrow \epsilon} + X_C \Delta G_C^{\gamma \rightarrow \epsilon} + X_{Nb}\Delta G_{Nb}^{\gamma \rightarrow \epsilon} \\
&+ X_{Fe}X_{Mn}\Omega_{FeMn}^{\gamma \rightarrow \varepsilon} + X_{Fe}X_{Al}\Omega_{FeAl}^{\gamma \rightarrow \varepsilon} + X_{Fe}X_C\Omega_{FeC}^{\gamma \rightarrow \varepsilon} + X_{Fe}X_{Nb}\Omega_{FeNb}^{\gamma \rightarrow \varepsilon} \\
&+ X_{Mn}X_C\Omega_{MnC}^{\gamma \rightarrow \varepsilon} + \Delta G_{mg}^{\gamma \rightarrow \epsilon}.
\end{aligned}
\tag{7}
$$

Based on Equation (1) and Equation (7), the values and functions used in the calculation are listed in Table 2 [26] and the $\Delta G^{\gamma \rightarrow \varepsilon}$ and SFE of steel at room temperature were estimated as 1128 J/mol and 84 MJ/m$^2$, respectively.

**Table 2.** The values used for the estimation of the stacking fault energy (SFE) of Fe–28Mn–10Al–C–0.5Nb steel by Equations (1), (4), and (7) [26].

| Parameters | Values and Functions |
|---|---|
| $\rho$ | $2.94 \times 10^{-5}$ (mol/m$^2$) |
| $\sigma$ | 9 (MJ/mol) |
| $\Delta G_{Fe}^{\gamma \to \epsilon}$ | $-2243.38 + 4.309T$ (J/mol) |
| $\Delta G_{Mn}^{\gamma \to \epsilon}$ | $-1000 + 1.123T$ (J/mol) |
| $\Delta G_{Al}^{\gamma \to \epsilon}$ | $2800 + 5T$ (J/mol) |
| $\Delta G_{C}^{\gamma \to \epsilon}$ | $-22{,}166$ (J/mol) |
| $\Delta G_{Nb}^{\gamma \to \epsilon}$ | $4046$ (J/mol) |
| $\Omega_{FeMn}^{\gamma \to \epsilon}$ | $2180 + 532(X_{Fe} - X_{Mn})$ (J/mol) |
| $\Omega_{FeAl}^{\gamma \to \epsilon}$ | $3339$ (J/mol) |
| $\Omega_{FeC}^{\gamma \to \epsilon}$ | $42{,}500$ (J/mol) |
| $\Omega_{MnC}^{\gamma \to \epsilon}$ | $26{,}910$ (J/mol) |
| $\Omega_{FeNb}^{\gamma \to \epsilon}$ | $27{,}403$ (J/mol) |
| $\beta^{\gamma}$ | $0.7X_{Fe} + 0.62X_{Mn} - 0.64X_{Fe}X_{Mn} - 4X_{C}$ |
| $\beta^{\epsilon}$ | $0.62X_{Mn} - 4X_{c}$ |
| $T_{nell}^{\gamma}$ | $580X_{Mn}(K)$ |
| $T_{nell}^{\epsilon}$ | $250\ln X_{Mn} - 4750X_{Mn}X_{C} - 6.2X_{Al} + 720(K)$ |
| $f\left(\dfrac{T}{T_{neel}}\right)$ | $1 - \dfrac{\left\{\left(\frac{79\tau^{-1}}{140\rho}\right) + \left(\frac{474}{497}\right)\left(\frac{1}{\rho}\right) - 1\right)\left(\left(\frac{\sigma^{3}}{6}\right) + \left(\frac{\tau^{6}}{135}\right) + \left(\frac{\tau^{15}}{600}\right)\right\}}{D}$ When $\tau = \dfrac{T}{T_{Nell}} < 1$ $1 - \dfrac{\left\{\left(\frac{\tau^{-5}}{10}\right) + \left(\frac{\sigma^{-15}}{315}\right) + \left(\frac{\tau^{-25}}{1500}\right)\right\}}{D}$ When $\tau = \dfrac{T}{T_{Nell}} > 1$. Where $\rho = 0.28, D = 2.34$ |

## 3.3. Mechanical Properties

The engineering stress-strain curve of the steel after solution treatment is shown in Figure 7. The steel exhibits continuous strain hardening during the tensile process at room temperature. The yield strength (YS) and ultimate tensile strength (UTS) of the sample were 963 MPa and 1084 MPa, respectively, and high elongation (EI) to fracture of 37.5% was obtained, where the elongation is the percentage ratio of the total deformation of gauge length after tensile fracture to the original gauge length. It is worth noting that compared with the mechanical properties of several typical Fe–Mn–Al–C low-density steels (Table 3), the present steel had higher YS and UTS. Although Fe–12Mn–5.5Al–0.7C had extremely high YS (1290 MPa) and UTS (1415 MPa), the EI of the steel was only 8.2% due to the existence of coarse intergranular κ-carbides, as shown in Figure 8, which makes it difficult to meet the requirements of commercial automotive steel [27]. This also indicates that ultimate tensile strength and ductility are mutually contradictory. Therefore, the product of strength and plastic (PSE = UTS × EI) is usually used to describe the balance between ductility and strength. Generally, a typical value of the PSE of TRIP steel and TWIP steel ranges from 15,000 to 20,000 MPa%, and as shown in Table 3, the range of the SFE of commercial Fe–Mn–Al-C low-density steel was about 11,000 MPa% to 88,000 MPa%. The PSE value of the present steel reached as high as 40,650 MPa%, much higher than the TRIP steel, TWIP steel, and traditional Fe–Mn–Al–C low density steel. The balance between ultra-high UTS (1084 MPa) and good ductility (37.5%) was achieved, which can meet the production and application requirements of automobile steel. The steel achieved a balance between ultra-high ultimate tensile strength (1084 MPa) and excellent ductility (37.5%), which satisfies the requirement for high performance automotive steels.

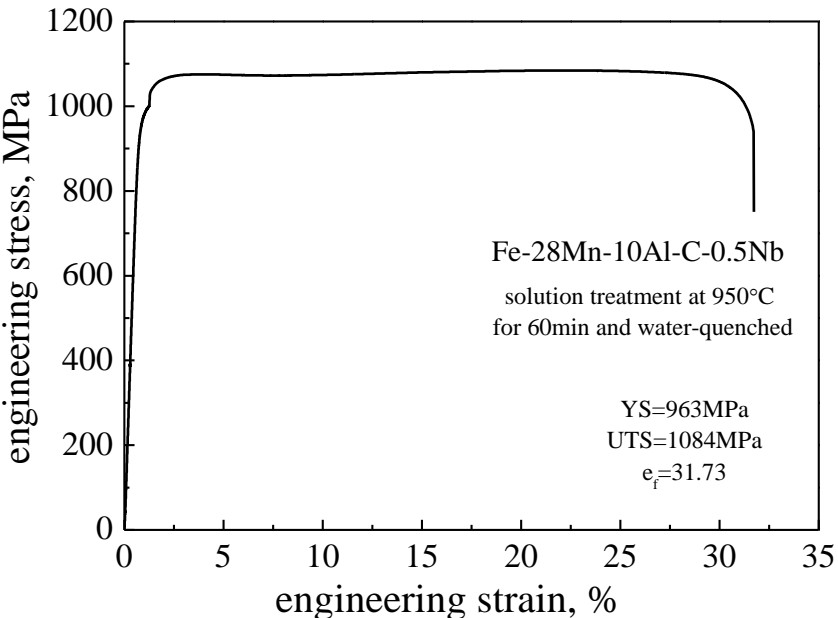

**Figure 7.** Engineering stress-strain curves of Fe–28Mn–10Al–C–0.5Nb low-density steel tested at room temperature at an initial strain rate of $1 \times 10^{-3} \mathrm{s}^{-1}$.

**Table 3.** Chemical compositions, annealing and cooling conditions, and mechanical properties of advanced high strength Fe–Mn–Al–C-based steels.

| Composition | Annealing and Cooling Condition | YS (MPa) | UTS (MPa) | EI (%) | PSE (GPa%) |
|---|---|---|---|---|---|
| Fe–28Mn–10Al–C–0.5Nb | 950 °C/60 min-water-quenched | 963 | 1084 | 37.5 | 40.65 |
| Fe–28Mn–9Al–0.8C [26] | 1000 °C/60 min-water-quenched | 440 | 880 | 100 | 88.00 |
| Fe–8.5Mn–5.6Al–0.3C [27] | 900 °C/30 min-air cooling | 502 | 734 | 77 | 56.52 |
| Fe–3.5Mn–5.8Al–0.35C [28] | 830 °C/15 s-air cooling | 622 | 800 | 42.0 | 33.60 |
| Fe–12Mn–5.5Al–0.7C [29] | 640 °C/10 min-air cooling | 1290 | 1415 | 8.2 | 11.60 |
| Fe–18Mn–10Al–1.2C [30] | 1000 °C/15 min-water-quenched | 702 | 875 | 77.4 | 67.72 |
| Fe–27Mn–12Al–0.9C [31] | 1025 °C/25 min-water-quenched | – | 875 | 58 | 50.75 |
| Fe–28Mn–10Al–C [31] | 1000 °C/60 min-water-quenched | – | 873 | 98.9 | 86.33 |
| Fe–30Mn–8Al–1.2C [31] | 1100 °C/120 min-water-quenched | – | 900 | 68 | 61.20 |
| Fe–26Mn–8Al–C [32] | 1000 °C/15 min-water-quenched | 625 | 915 | 50.9 | 46.57 |

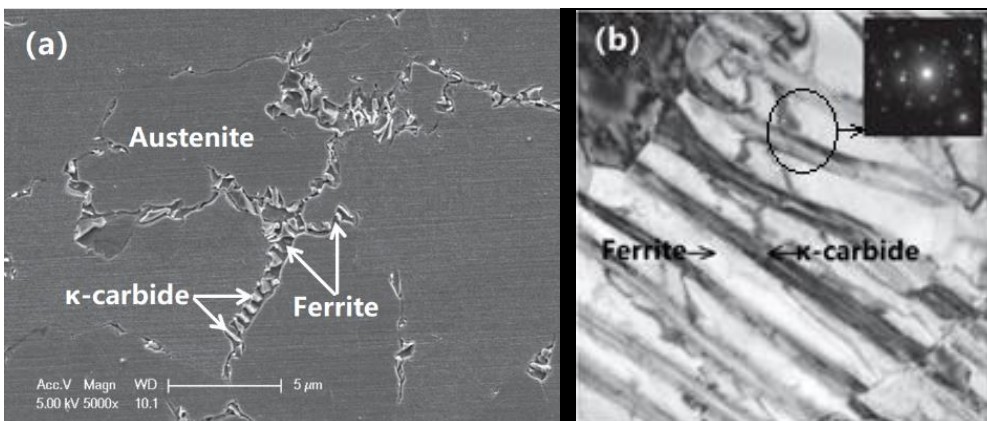

**Figure 8.** (**a**) SEM and (**b**) TEM micrographs of the Fe–12Mn–5.5Al–0.7C steel, showing ferrites and κ-carbides at the boundaries of austenite grains in [27].

The true stress (σ)-true strain (ε) curve of the steel is presented in Figure 9 with the corresponding strain hardening rate (dσ/dε). It can be seen from the results that

the strain hardening rate was not linear; its rate rapid decreased in the initial stage of plastic deformation, then continuously increased to $\varepsilon = 0.175$, subsequently $d\sigma/d\varepsilon$ began to decrease to $\varepsilon = 0.24$ and the occurrence of plastic instability. The post-necking strain was 0.31%.

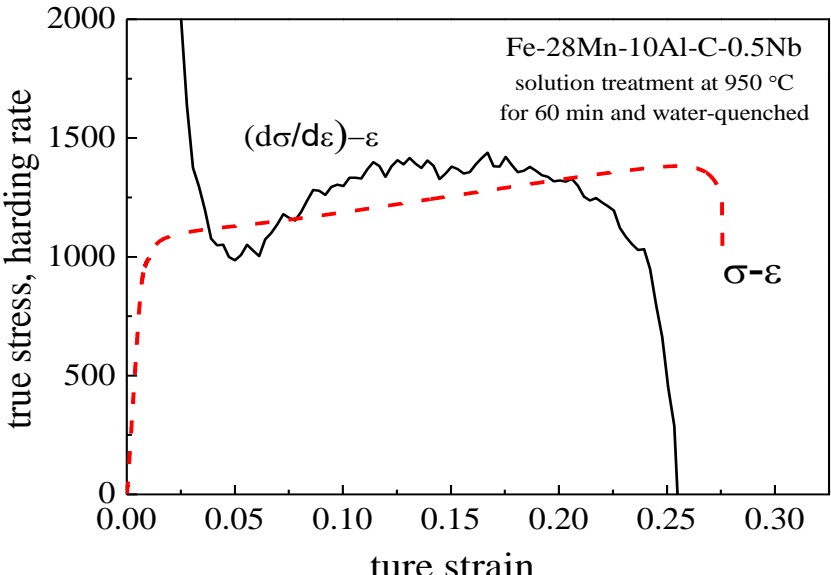

**Figure 9.** True stress-strain curves of Fe–28Mn–10Al–C–0.5Nb low-density steels and the corresponding strain hardening rate as a function of true strain.

### 3.4. Deformed Micro-Structures

In order to further determine the mechanism of plastic deformation of the present steel, dislocation configuration of the sample was performed through intermittent tensile tests under different strain levels of 0.05, 0.15, and 0.25 and the failure samples, as shown in Figure 10. As shown in Figure 10a, at a low strain level of 5%, the development of the substructure manifested through dislocation pile ups on a single slip plane and the slip along the {1 1 1} plane, which are typical planar slip configurations. When the strain increased to 10%, no distinct cell structure was found, and the dislocation structure still showed a planar slip configuration. Moreover, the slip trace on another plane was observed, which indicates an activation of multiple slip with further increment, and a Taylor lattice-like structure, a kind of low-energy dislocation structure, was formed, as shown in Figure 10b. As the sample strained to a medium strain of 25%, microbands were exhibited with distinct boundaries (as marked 'A' in Figure 10c). With the increase in strain, the spacing between the slip bands further reduced, and the microband structure (marked "B" in Figure 10c) appeared. When the sample failed to fracture, as marked 'A' in Figure 10d, the slip became finer and more intensive. As more multiple slips occurred, the sub microstructure of the sample was dominated by the intersections of microbands. In addition, the substructure of the fracture sample still showed a planar slip structure, and no dislocation cell was formed.

To investigate the fracture mode of Fe–Mn–Al–C–Nb low-density steel, the fracture morphology after tensile fracture was observed by SEM, as shown in Figure 11. The fracture surface was composed of well-developed dimples [33,34], indicating that the fracture mode was ductile fracture [35,36].

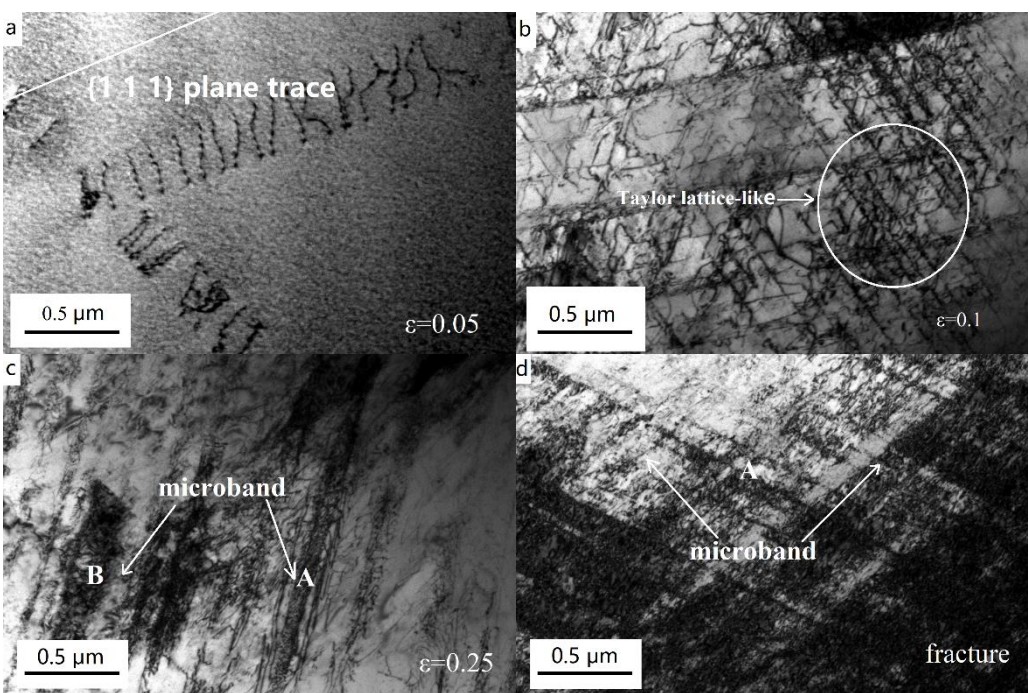

**Figure 10.** TEM micrographs of the Fe–28Mn–10Al–C–0.5Nb low-density steel after interrupted at different strains during room temperature tensile deformation of (**a**) 5%, (**b**) 10%, (**c**) 25%, and (**d**) failure to fracture.

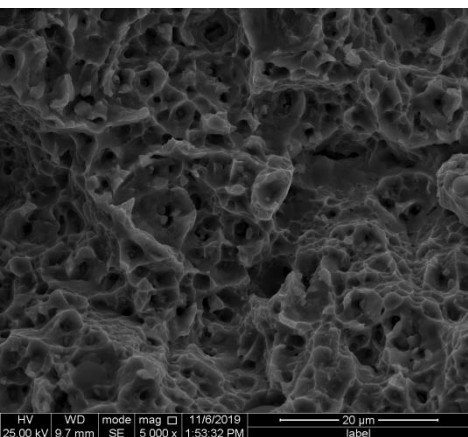

**Figure 11.** SEM morphology of dimpled ductile fracture surface of Fe–28Mn–10Al–C–0.5Nb low-density steel.

## 4. Discussion

The analysis of the above results showed that the present Fe–28Mn–10Al–C–0.5Nb low-density steel had high ultimate tensile strength and elongation after fracture. It showed continuous strain hardening during room temperature tensile, and the deformed microstructure exhibited typical planar slip characteristics such as the formation of dislocation pile-up, Taylor lattice, and microband, while no cell formed, nor were martensite and mechanical twins produced. This is mainly attributed to the influence of SFE on the deformation mode. According to the previous work, the TRIP effect is dominant during the deformation of austenite when the SFE is less than 18 MJ/m$^2$, and the deformation twins will replace the martensitic transformation when SFE is between 18 and 40 MJ/m$^2$L the deformation of austenite is manifested by TWIP effect [37]. The SFE (84 MJ/m$^2$) of the present steel was much higher than that of the TWIP effect. Accordingly, it is instructive to describe the effect of SFE on the deformation mode of austenitic low-density steels.

### 4.1. Stacking Fault Energy and Deformation Modes of Austenitic Steel

It can be judged by the Olson–Cohen mode ($\tau = 2\rho\Delta G^{\gamma\rightarrow\varepsilon} + 2\sigma$) that when the SFE of austenite is less than the $\gamma/\varepsilon$ interface energy, $\Delta G^{\gamma\rightarrow\varepsilon}$ becomes negative. In this case, the overlapping of the intrinsic stacking faults on the {111} plane will promote the formation of new planar defects-shear bands during the plastic deformation of austenite. With further deformation, the number of shear bands increase and the $\varepsilon$ martensite with HCP structure is formed at their intersections for the reason of low energy, causing the transformation-induced plasticity [26]. Generally, the typical value of $\gamma/\varepsilon$ interface energy is 10–20 MJ/mol. Accordingly, based on the calculation results of Equation (1), as the SFE was less than 20 MJ/m$^2$ in the FCC solid solution alloys [26], the value of $\Delta G^{\gamma\rightarrow\varepsilon}$ was negative, and the deformed microstructure exhibited the increasing $\alpha'$-martensite volume fraction with increasing strain (i.e., the plastic deformation mode is the TRIP effect).

In contrast, when SFE is higher than the $\gamma/\varepsilon$ interfacial energy, $\Delta G^{\gamma\rightarrow\varepsilon}$ is a positive value, which means that the formation of $\varepsilon$-martensite will be inhibited during the strain process. In addition, due to the continuous displacement of partial dislocations on the slip surface, the formation of mechanical twinning is favorable, as a result, strain hardening of austenitic steel will be enhanced (i.e., the TWIP effect). According to previous work, based on the relationship between partial dislocation and stress, the critical stress $\sigma_T$ for the occurrence of twinning is summarized as [38]:

$$\sigma_T = 6.14\frac{\tau}{b} \tag{8}$$

where $b$ is the magnitude of the partial dislocation and the value is usually 0.147 nm [39]. As the applied load is larger than the critical stress ($\sigma_T$), the stability of the stacking fault decreases and diverges into partial dislocations. Therefore, twin stacking faults will be formed by continuous decomposition of stacking faults, and lead to the formation of mechanical twins. Inversely, while the applied load is lower than $\sigma_T$, no mechanical twins are formed during the deformation process. Instead, deformation is achieved by dislocation gliding.

According to the previous calculation, the SFE of Fe–28Mn–10Al–Nb steel was much higher than the value of the $\gamma/\varepsilon$ interface energy, which will suppress the formation of shear bands and martensitic transformation during deformation. Based on Equation (8), the $\sigma_T$ of the present Fe–Mn–Al–C–Nb low-density steel was estimated to be as high as 3429 MPa by using $\tau = 84$ MJ/m$^2$, which is much higher than its UTS. Therefore, it can be concluded that no $\varepsilon$ martensite nor mechanical twins occur during tensile deformation at room temperature. Instead, plastic deformation of Fe–Mn–Al–C–Nb low-density is realized by planar slip, as observed by TEM.

### 4.2. Glide Plane Softening and Microband-Induced Plasticity

#### 4.2.1. Glide Plane Softening

Generally, for austenite materials, SFE not only affects the deformation mechanism, but also determines the slip mode of dislocations in the process of plastic deformation [40]. When the SFE is low, the two partial dislocations decomposed by perfect dislocations are easy to separate and have strong mobility, therefore Burgers vector directions for cell formation are insufficient, tending to planar slip, as shown in Figure 12a [41]. As the SFE is high enough, the width of the extended dislocation decreases. During the progress of plastic deformation, the cross-slip of extended partials can easily occur, which promotes the wavy slip and the formation of a cell structure, as shown in Figure 12b. However, the present low-density steel still exhibited a planar slip structure, even though it had a high SFE (84 MJ/m$^2$), while no wavy slip and cell formation occurred, even up to failure. These results indicate that the SFE is not the only parameter that determines dislocation slip mode (i.e., planar slip or wavy slip).

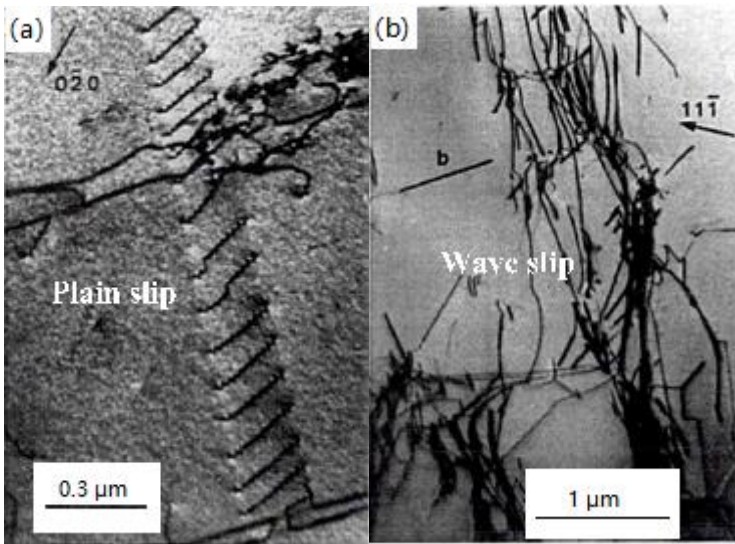

**Figure 12.** Dislocation plane slip structure (**a**) and wavy slip (**b**) [40].

According to several previous reports, regardless of the value of SFE, the formation of short range ordering (SRO) in austenitic steels also determines the mode of dislocation slip [42]. In the process of tensile deformation at room temperature, once the leading dislocation shears the short range ordering, the structure of SRO will be destroyed and it is difficult to self-recover. Therefore, the slip resistance of dislocations in this region decreases, and it will facilitate propagation of succeeding dislocations on the same glide plane with high slip rate, forming a slip plane. Thus, the gliding of dislocations more easily gets through the destroyed ordered region on the same plane. The softening phenomenon is referred to as 'glide plane softening' [43,44]. Accordingly, dislocations were exhibited as the planar glide manner on the slip plane in the process of plastic deformation. It is reported that in concentrated solid solutions, planar slip caused by glide plane softening occurs more preferentially [26]. The Fe–28Mn–10Al–C–0.5Nb low-density steel in the solid solution state is also a concentrated solid solution with the total atomic mole fraction of the alloying elements of 0.47. Therefore, due to its relatively high SFE and concentrated alloying, the plane slip of the dislocations in the present steel during the plastic deformation can be attributed to the glide plane softening effect, rather than the SFE effect.

Additionally, Figure 13 shows the relationship between dislocations and precipitates at 5% deformation of the present steel. As can be seen from Figure 13, the planar slip of dislocation will shear the precipitates (NbC) the present steel during tensile deformation at room temperature. Due to the hindering of precipitation (NbC), the leading dislocation moving through the slip plane faces the energy barrier by NbC, so the following dislocations are piled up on the same single slip plane, and the plane slip is blocked. The leading dislocation overcomes the NbC energy barrier by shearing and destroying the structure of NbC by itself. Since the destroyed precipitation is hard to restore, it will facilitate the following dislocation to propagate on the same glide plane, and promote the planar slip of dislocation and the form of planar slip bands. The result indicates that similar to SRO, NbC will be crystallographically sheared, which is considered as weak obstacles to the movement of dislocations [16], hence promoting the glide plane softening and the planar gliding. Moreover, the precipitation of NbC did not alter the planar slip mode in the process of tensile deformation at room temperature.

### 4.2.2. Microband-Induced Plasticity

As a well-recognized deformation mechanism of high SFE materials, planar slip was first proposed by Frommeyer and Brüx [31]. In the process of the mechanical behavior of Fe–27Mn–12Al–0.9C low-density steel, it was found that austenite, ferrite, and κ-carbide coexisted in the low-density steel. In the case of high SFE (110 MJ/m$^2$), Fe–27Mn–12Al–

0.9C steel obtained high UTS (875MPa) and EI (58%) during the tensile deformation at room temperature. Based on the TEM observation, researchers believe that high strength and ductility are mainly due to the formation and existence of κ-carbides. During the process of plastic deformation, dislocation motion shears through κ-carbide and forms shear bands, then induces Fe–27Mn–12Al–0.9C low-density steel to obtain high plasticity (i.e., shear band induced plasticity (SIP)). Yoo et al. [26] found that Fe–28Mn–9Al–0.8C had a high UTS (840 MPa) and an incredibly high EI (100%) at room temperature. Observed by TEM and fracture morphology, there was no TRIP or TWIP effect in the process of tensile deformation, nor were the shear bands observed by Frommeyer and Brüx. Instead, the microbands and intersections of well-developed microbands were formed in the process of tensile deformation. The excellent combination of strength and ductility of the steel can be primarily attributed to microband-induced plasticity.

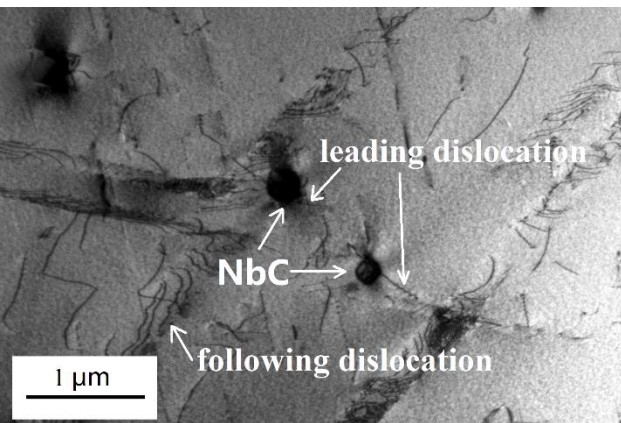

**Figure 13.** Interaction between dislocations and precipitates in 0.5Nb low density steel at 5% strain.

As mentioned in Section 3.4, in the initial stage of deformation, the structure of dislocation was narrow and dislocations had high three-dimensional mobility. With increasing strain, dislocations slip along the slip plane. Due to the alternating sense, the density of dislocations gradually increases and the Taylor lattices are formed. Moreover, with the activation of the non-coplanar slip system, the submicroscopic of the sample will be dominated by the intersections of well-developed microbands. It can be seen that under the process of strain, the submicroscopic evolution process of the present Fe–Mn–Al–C–Nb low-density steel can be described as dislocation pile-up → Taylor lattice → intersections of microbands. Moreover, no shear band formation was found during the deformation process. In addition, since both the Taylor lattice and the microbands are the structure consisting of geometrically necessary dislocations, the total dislocation density gradually increases as the strain increases, and therefore continuous strain hardening occurs. Furthermore, the formation and intersection of microbands will act as sub-grain boundaries and penetrate the austenite grains. On one hand, it can hinder planer slip and strengthen the steel; on the other hand, as an area of stress concentration, new dislocations can form near the microbands, improving the strain hardening rate, and increasing the strength and toughness of steel at the same time. Accordingly, similar to the conclusions of Yoo et al. [26], the exceedingly good balance between high strength and ductility of the present Fe–Mn–Al–C–Nb low-density steel can also be attributed to the microband-induced plasticity, rather than the shear band induced plasticity proposed by Frommeyer and Brüx.

In addition, it can be seen from Figure 13 that the precipitations (NbC) in Fe–Mn–Al–C–Nb low-density steel will hinder the planar slip of dislocations, which will enhance the precipitation strengthening effect and improve the strain hardening rate, so the present steel will obtain higher YS and UTS than the same type of low density steel without the addition of Nb. This indicates that the addition of Nb does not change the deformation mechanism and strengthening mechanism of Fe–Mn–Al–C low-density steel, and can optimize the mechanical properties of the steel.

Based on the practical perspective, through Al alloying and utilizing microband-induced plasticity, Fe–Mn–Al–C–Nb steel has low-density and competitive mechanical properties, which meets the requirements of high-performance automotive steels and can be achieved for wide use in the future.

## 5. Conclusions

1. Fe–28Mn–10Al–C–0.5Nb steel had a low density (6.63 g/cm$^3$) after solution treatment, and was fully austenitic with extremely few annealing twins. The $\Delta G^{\gamma \rightarrow \varepsilon}$ and stacking fault energy of the steel at room temperature was estimated to be 1128 J/mol and 84 MJ/m$^2$, respectively.
2. The fully austenitic Fe–28Mn–10Al–C–0.5Nb low-density steel showed an excellent ultimate tensile strength (1084 MPa) and elongation (37.5%), and the steel exhibited an excellent combination of strength and ductility with the product of strength and plastic value of 40.65 GPa%.
3. The Fe–28Mn–10Al–C–0.5Nb low-density steel exhibited typical planar glide characteristics during deformation. The excellent combination of strength and ductility can be attributed to plasticity induced by microbands and leads to the continuous strain hardening during deformation at room temperature.
4. The addition of Nb did not change the deformation mechanism and strengthening mechanism of Fe–Mn–Al–C low-density steel, and in the absence of aging treatment and without the formation of κ-carbide, the present Fe–Mn–Al–C–Nb steel achieved a balance between ultra-high ultimate tensile strength and excellent ductility, which can optimize the mechanical properties of the steel.

**Author Contributions:** Conceptualization, T.M. and Y.L.; methodology, T.M., J.G. and H.L.; writing—original draft preparation, T.M., H.Z. and C.L.; writing—review and editing, J.G. and Y.L. formal analysis and validation, T.M., J.G. and Y.L.; project administration, Y.L.; funding acquisition, Y.L.; and supervision, Y.L. All authors have read and agreed to the published version of the manuscript.

**Funding:** This research was funded by National Natural Science Foundation of China grant number 51974129.

**Institutional Review Board Statement:** Not applicable.

**Informed Consent Statement:** Not applicable.

**Data Availability Statement:** Data is contained within the article.

**Acknowledgments:** This work was financially supported by grants through the National Natural Science Foundation of China (Nos. 51974129).

**Conflicts of Interest:** The authors declare no conflict of interest.

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
