# Peer review of "Microband-Induced Plasticity in a Nb Content Fe–28Mn–10Al–C Low Density Steel"

_metals, doi:10.3390/met11020345_

Round 1

Reviewer 1 Report

The authors studied the mechanical properties of Fe-Mn-Al-Nb-C steel. The results can be useful for the researchers in this field, but the paper has to be revised before publication.

1. p. 5, line 12, 13
"G・Pa%" should be "GPa%."
Please consider to use the description like "MPa%."

2. p. 5, line 14
"mpa" should be "MPa."

3. p. 8, line 3
The crystal structure of alpha' is BCC.
Please confirm this.

4. p. 9, Figure 8
These pictures do not seem to be present results.
If they are taken from some literature, it should be shown in reference section.

5. p. 10, Figure 9
It seems that the size and distribution of NbC particles can not account for the planar slip observed in the present study. The planar slip in similar steels has been reported in previous papers where the sample has dense distribution of fine carbide particles. Here, the authors should show the reason of the planar slip other than the existence of NbC.

Author Response

Point 1: p.5,line 12,13 "G·Pa%" should be "GPa%." Please consider to use the description like "MPa%."

Response 1: The unit has been described as “MPa%” as required

Point 2: p.5, line 14 "mpa" should be "MPa."

Response 2: The unit has been described as “MPa%” as required to replace “mpa”

Point 3: p. 8, line 3 The crystal structure of alpha' is BCC.Please confirm this.

Response 3: Due to typos, “ε martensitic” was mistakenly written as “α martensitic”, which has been corrected.

Point 4: p.9, Figure8 These pictures do not seem to be present results. If they are taken from some literature, it should be shown in reference section.

Response 4: Figure 8 is indeed taken from other literature, which has been marked and shown in the reference section

Point 5: p.10, Figure 9. It seems that the size and distribution of NbC particles can not account for the planar slip observed in the present study. The planar slip in similar steels has been reported in previous papers where the sample has dense distribution of fine carbide particles. Here, the authors should show the reason of the planar slip other than the existence of NbC.

Response 5: The sample has a dense distribution of fine carbide particles, resulting in a plane slip in similar steels by ‘glide plane softening’. In the present study, due to the control of element content and heat treatment temperature, the precipitation of fine carbide particles in steel is avoided. However, because of the precipitation of NbC, the planar slip of dislocation will shears the precipitates (NbC) of the present steel during tensile deformation at room temperature, resulting in the phenomenon of ‘glide plane softening’. Accordingly, dislocations exhibits as the planar glide manner on the slip plane in the process of plastic deformation. So I think that the size and distribution of NbC particles could account for the planar slip observed in the present study in Fig.10 (Fig.9 in the original article).

Reviewer 2 Report

1. Unfortunately, the tensile tests were carried out by the authors on samples with non-uniform geometry and not in accordance with the ASTM E-8 standard (50 mm 12.5 mm 1 mm) (see Fig. 1). The standard rate of relative deformation of the working part of the sample, if possible, should be maintained constant and equal to 10+3 1/sec. The authors used a deformation rate of 10-3 1/sec (see Fig. 3). Usually, this rate is used by the authors to transfer the metal to a state of superplasticity.
2. According to the tensile curve for the test steel Fe–28Mn–10Al–1.0C–0.5 Nb (see Fig. 3), neck formation and fracture occurred at a strain of 30.2% (true strain of 0.25) (see Fig. 3, Fig. 4). While, in the review article (Shangping Chen, Radhakanta Rana, Arunansu Haldar, Ranjit Kumar Ray. Current state of Fe-Mn-Al-C low density steels. Progress in Materials Science. Volume 89, August 2017, Page 22. Pages 345-391. https://doi.org/10.1016/j.pmatsci.2017.05.002) for Fe–28Mn–10Al–1.0C steel, fracture occurred at 90% strain (true strain 0.67) (see Page 23).
3. For electron microscopic studies, the authors deformed the samples at a rate of 10-3 1/sec to a true strain of 0.05, 0.1, 0.25 and complete destruction. I would like to get acquainted with the electron microscopic structure of Fe–28Mn–10Al–1.0 C–0.5 Nb steel after the deformation rate 10+3 1/sec.

Author Response

English language and style have been checked and improved as required. 

Reviewer 3 Report

The manuscript entitled: Microband-induced plasticity in a Nb contented Fe-28Mn–10Al–C low density steel 

  • Chemical composition should be introduced in the form of a table with errorbars.
  • The heat treatment cycle may be well represented in a pictorial format.
  • How can the density of twins (low twins density) be ascertained from the optical image in Fig. 2?
  • Authors have claimed that the higher strength is due to the presence of the existence of coarse intergranular κ-carbides. However, the proof is missing. Either HRSEM and /or TEM images should be introduced to show the coarse intergranular κ-carbides.
  • XRD diffraction patterns should be introduced to show the presence of the phases.
  • Authors have claimed that: 'During the straining process, stacking faults will be formed on the {111} plane of the FCC structure,and the perfect dislocations will decompose into partial dislocations, and the intrinsic stacking fault is formed on the {1 1 1} plane, due to the movement of partial dislocations, as marked by the dashed line in Fig. 7'. Proof of the same is missing. Firstly, the presence of stacking faults and decomposition of partial dislocations should be proved experimentally.
  • The authors have a strong scientific discussion. However, they do not accompany with sufficient experimental proof. 
  • Typos in the manuscript need to be rectified. For instance, 25kg should be written as 25 kg (space between number and unit should be introduced).
  • The English language needs attention. 

Author Response

Point 1: Chemical composition should be introduced in the form of a table with errorbars.

Response 1: The composition of present steel has been introduced in the form of a table with errorbars, as shown in “Table 1. Chemical compositions of designed steels” in the present work.

Point 2: The heat treatment cycle may be well represented in a pictorial format

Response 2: The heat treatment cycle has been represented in a pictorial format, as shown in Figure 1 in the present article.

Point 3: How can the density of twins (low twins density) be ascertained from the optical image in Fig. 2?

Response 3: Through repeated inspection and confirmation of the experimental results, as suggested by the reviewer’s comments, the density of twins cannot be ascertained by from the optical image in Fig. 2, this part of the conclusion has been deleted from the original article.

Point 4: Authors have claimed that the higher strength is due to the presence of the existence of coarse intergranular κ-carbides. However, the proof is missing. Either HRSEM and /or TEM images should be introduced to show the coarse intergranular κ-carbides

Response 4: As requested, the SEM and TEM images has been introduced to show the coarse intergranular κ-carbides, as shown in Fig.5 in the new version of my manuscript.

Point 5: XRD diffraction patterns should be introduced to show the presence of the phases.

Response 5: XRD diffraction pattern has been introduced to show the presence of the phases in the new version of my manuscript, indicating the presence face-centered cubic (fcc/ bcc) peaks and the sample was fully austenitic.

Point 6: Authors have claimed that: 'During the straining process, stacking faults will be formed on the {111} plane of the FCC structure,and the perfect dislocations will decompose into partial dislocations, and the intrinsic stacking fault is formed on the {1 1 1} plane, due to the movement of partial dislocations, as marked by the dashed line in Fig. 7'. Proof of the same is missing. Firstly, the presence of stacking faults and decomposition of partial dislocations should be proved experimentally.

The authors have a strong scientific discussion. However, they do not accompany with sufficient experimental proof.

Response 6: This part shown in the manuscript are derived from the conclusions of other people's literature to describe the relationship between stacking faults and dislocations. Unfortunately, our work has not verified this. According to the reviewer’s comments, we conducted a detailed discussion and found that this part of the content has little relevance to describing the stacking fault energy and deformation mechanism, and it is not accompany with sufficient experimental proof, such as the presence of stacking faults and decomposition of partial dislocations. Therefore, in order to avoid this obvious loophole, I have deleted this part of the description without affecting the discussion of the experimental results.

Point 7: Typos in the manuscript need to be rectified. For instance, 25kg should be written as 25 kg (space between number and unit should be introduced).

Response 7: Rectified carefully for typos in the manuscript, such as 25kg has been written as 25 kg.

Point 8: The English language needs attention

Response 8: Check and modify English language and spelling

Reviewer 4 Report

The authors present a potentially interesting work. They use analysis tools that have great potential in materials characterization. However, the design of the research is very poor in one particular aspect: they intend to analyse the effect of Nb on the resulting alloy, but they just consider one Nb content. As a result they obtain conclusions that are not demonstrated to be related with Nb, and could be related to any other variable in the process. In other words, the provide opinions more than scientific conclusions.

Other minor comments (just a few of a number):

  • Page1: “However, due to the rise of fuel costs and the increasing air pollution, automobile manufacturers are committed to developing vehicles with low fuel consumption and high safety. However, due to rising fuel costs and restrictions on exhaust emissions, automakers have begun to devote to developing vehicles with low fuel consumption and high safety. “

Authors repeat basically the same sentence.

  • Page 1: “Therefore, it is widely concerned around the world [5-7].”

This sentence has not complete meaning.

  • Page 2: “…will precipitated..”

Typo.

  • Page 2: the whole last paragraph of Section 1 has a lot of statements concerning how Nb affects final properties. This paragraph requires references each time a significant statement is included.
  • Page 2, first paragraph of Section 2: it is described the fabrication process, but the aim of the different steps should be briefly indicated. Also, units of chemical composition should be included (e.g. wt.%).
  • Did the authors follow any tensile testing standard. Please, indicate.
  • Section 3.1: “This indicates that the growth of austenite grains and the formation of twins will be inhibited by the addition of Nb.”. This is not necessarily true, as Nb is not the only change between the material being analysed and that analysed in [31].
  • Table 1. It is not sufficiently justified why such values may be taken in this research.
  • Section 3.3. The value of ductility equal to 37.5% does not match with the graph shown in Figure 3
  • Section 3.4: “indicating that fracture mode is ductile fracture rather than brittle fracture caused by shear band”.

This sentence is very confusing concerning fracture mechanisms in brittle conditions (cleaveage, mainly).

Author Response

Point 1: The authors present a potentially interesting work. They use analysis tools that have great potential in materials characterization. However, the design of the research is very poor in one particular aspect: they intend to analyse the effect of Nb on the resulting alloy, but they just consider one Nb content. As a result they obtain conclusions that are not demonstrated to be related with Nb, and could be related to any other variable in the process. In other words, the provide opinions more than scientific conclusions. 

Response 1: In this work, the main purpose is to study the deformation mechanism of Fe-Mn-Al-C low-density steel with Nb added in the room temperature tensile deformation process and the corresponding reasons. The final conclusion also proves the excellent combination of strength and ductility is attributed to plasticity induced by microbands and leads to the continuous strain hardening during deformation at room temperature. Moreover, by comparing other steels under similar conditions, the addition of Nb does not change the deformation mechanism and strengthening mechanism of Fe-Mn-Al-C low-density steel, and can optimize the mechanical properties of the steel. Therefore, the design of the research just consider one Nb content.

Point 2: Page1: “However, due to the rise of fuel costs and the increasing air pollution, automobile manufacturers are committed to developing vehicles with low fuel consumption and high safety. However, due to rising fuel costs and restrictions on exhaust emissions, automakers have begun to devote to developing vehicles with low fuel consumption and high safety. “

Authors repeat basically the same sentence.

Response 2: The basically  same sentence” However, due to the rise of fuel costs and the increasing air pollution, automobile manufacturers are committed to developing vehicles with low fuel consumption and high safety” has been deleted.

Point 3: Page 1: “Therefore, it is widely concerned around the world [5-7].”

This sentence has not complete meaning.

Response 3: This sentence has been completed as “Therefore, the development of Fe-Mn-Al-C low-density steels are widely concerned around the world [5-7].”

Point 4: Page 2: “…will precipitated..”

Typo.

Response 4: The typo has been revised to precipitate.

Point 5: Page 2: the whole last paragraph of Section 1 has a lot of statements concerning how Nb affects final properties. This paragraph requires references each time a significant statement is included.

Response 5: As required, references has added to describe how Nb affects final properties in the whole last paragraph of Section 1.

Point 6: Page 2, first paragraph of Section 2: it is described the fabrication process, but the aim of the different steps should be briefly indicated. Also, units of chemical composition should be included (e.g. wt.%).

Response 6: The aim of the different steps has been briefly indicated as “In order to obtain a plate-like shape, the ingot is forged and hot rolled, and the high temperature heat treatment is performed before and after machining to remove internal stress. “. And the chemical composition of present steel has been introduced in the form of a table with units “wt.%”.

Point 7: Did the authors follow any tensile testing standard. Please, indicate.

Response 7:  Following the tensile testing standard of “GB/T 228.1-2010 sub-size standard”, and indicates in the new version of my manuscript.

Point 8: Section 3.1: “This indicates that the growth of austenite grains and the formation of twins will be inhibited by the addition of Nb.”. This is not necessarily true, as Nb is not the only change between the material being analysed and that analysed in [31].

Response 8:  As stated in the comments made by the reviewers, because of Nb is not the only change between the material being analysed and that analysed in [31], the conclusions presented in the manuscript “This indicates that the growth of austenite grains and the formation of twins will be inhibited by the addition of Nb." is an inaccurate conclusion. To avoid misstatement, deleted this conclusion in the new version of my manuscript.

Point 9: Table 1. It is not sufficiently justified why such values may be taken in this research..

Response 9: After the experimental verification and relevant calculation in reference [24], the accuracy of the values used in Table 1 can be sufficiently justified, and can be used in the calculation of this work. Table 1 has been added with references in the new version of my manuscript.to justified why such values may be taken in this research.

Point 10: Section 3.3. The value of ductility equal to 37.5% does not match with the graph shown in Figure 3

Response 10: Figure 3 shows the “engineering strain”, and the value of ductility equal to 37.5% is the elongation (EI) to fracture. And the EI is calculated based on the tensile testing standard of GB/T 228.1-2010.

Point 11: Section 3.4: “indicating that fracture mode is ductile fracture rather than brittle fracture caused by shear band”.

This sentence is very confusing concerning fracture mechanisms in brittle conditions (cleaveage, mainly).

Response 11: The original meaning of this sentence is to indicate that in the observation of fracture morphology, since only well-developed dimples are observed, but no shear bands are found, fracture mode is judged as ductile fracture. In the follow discussion, the reason why the shear band did not appear was also explained. But through the reviewer’s comments, we also found that this sentence is very confusing, so the conclusion “rather than brittle fracture caused by shear band “is deleted, and use a more concise way to describe the fracture mode

Reviewer 5 Report

  1. Please, complete the results confirming the presence of NbC carbides and k-carbides (diffraction pattern and its solution or EBSD)
  2. The structure is not sufficiently investigated both in the initial state and after tensile tests.No carbide analysis both NbC and k-carbides and their role during tensile tests.
  3. Clearly state the novelty aspects of the paper;
  4. The aim of the work is unclear: „In order to better improve the mechanical properties, the present work aims to develop a Nb alloyed Fe-Mn-Al-C low-density steel”. What specific parameters of mechanical properties are considered?
  5. The methodology for determining the average grain size should be described.
  6. Conclusions should highlight the new findings of the paper in comparision to the state of the art.
  7. There is a lot of flaws. The paper requires thorough checking. For instance: language style „In addition, the addition of Al…”-  „et al” instead of et al.
  • language style „In addition, the addition of Al…”
  • „et al” instead of et al.
  • Common language „As we all know…”
  • „Figure 3. engineering…” instead of „Figure 3. Engineering…”
  • „(1084mpa)” instead of (1084 MPa),
  • No spaces between parameters indications and units, for instance: YS(MPa) instead of YS (MPa)
  • No spaces between different sentences „hardening rate (dσ/dε).It can”
  • Style „there is no distinct cell structure was found…”

      Extensive English correction is needed

Author Response

Response to Reviewer 5 Comments

Point 1: Please, complete the results confirming the presence of NbC carbides and k-carbides (diffraction pattern and its solution or EBSD)

Response 1: The presence of NbC carbides  has been confirmed by SEM image and TEM bright-field morphologies and EDS analysis

Point 2: The structure is not sufficiently investigated both in the initial state and after tensile tests.No carbide analysis both NbC and k-carbides and their role during tensile tests.

Response 2: The carbide analysis of  NbC has been tested by SEM image and TEM bright-field morphologies and EDS analysis in the initial state tensile tests. And the interaction between dislocations and precipitates in 0.5Nb low density steel at 5% strain has shown in Fig.13 as required.

Point 3: Clearly state the novelty aspects of the paper;

Response 3: Modify as required, pointing out that novelty aspects of the paper  is “At present, there is less report about the preparation of Fe-Mn-Al-C-Nb low density steel by NB alloying, and the evolution of the microstructure of Fe-Mn-Al-C-Nb low density steel during plastic deformation has not attracted attention, and the strengthening mechanism is still unclear.”

Point 4: The aim of the work is unclear: „In order to better improve the mechanical properties, the present work aims to develop a Nb alloyed Fe-Mn-Al-C low-density steel”. What specific parameters of mechanical properties are considered?

Response 4: Modify as required, pointing out that specific parameters of mechanical properties is tensile property.

Point 5: The methodology for determining the average grain size should be described.

Response 5: Mentioned that “The average grain size was determined by Nano Measurer.”

Point 6: Conclusions should highlight the new findings of the paper in comparision to the state of the art.

Response 6: Pointed out new findings of the paper in comparision to the state of the art as “in the absence of aging treatment and without the formation of κ-carbide, the present Fe-Mn-Al-C-Nb steel achieves a balance between ultra-high ultimate tensile strength and excellent ductility, which can optimize the mechanical properties of the steel..”

Point 7: There is a lot of flaws. The paper requires thorough checking. For instance: language style „In addition, the addition of Al…”-  „et al” instead of et al.

language style „In addition, the addition of Al…”

„et al” instead of et al.

Common language „As we all know…”

„Figure 3. engineering…” instead of „Figure 3. Engineering…”

„(1084mpa)” instead of (1084 MPa),

No spaces between parameters indications and units, for instance: YS(MPa) instead of YS (MPa)

No spaces between different sentences „hardening rate (dσ/dε).It can”

Style „there is no distinct cell structure was found…”

      Extensive English correction is needed

Response 7: Have been modified as required and do the extensive English correction.

Round 2

Reviewer 1 Report

It still seems that the size and distribution of NbC particles can not account for the planar slip observed in the present study. If the authors insist that the NBC particles account for the planar slip, they should show direct evidence. For example, the deformation microstructure in the sample without NbC particles should be shown.

Author Response

Response to Reviewer 1 Comments

Point 1: It still seems that the size and distribution of NbC particles can not account for the planar slip observed in the present study. If the authors insist that the NBC particles account for the planar slip, they should show direct evidence. For example, the deformation microstructure in the sample without NbC particles should be shown.

Response 1: After discussion, the conclusion of this part was revised. We believe that the reason for the plane slip is due to the high alloying in the low-density steel. Similar to the previous study, the presence of SRO in the concentrated solid solutions will cause the glide plane softening, which will cause the plane slip. Fe-28Mn-10Al–C-0.5Nb low-density steel in the solid solution state is also a concentrated solid solution with the total atomic mole fraction of the alloying elements of 0.47. Therefore, due to its relatively high SFE and concentrated alloying, the plane slip of the dislocations in the present steel during the plastic deformation is attributed to the glide plane softening effect, rather than SFE effect. Additionally,NbC will be crystallographically sheared, which considered as weak obstacles to the movement of dislocations, hence promoting the glide plane softening and the planar gliding. Moreover, the precipitation of NbC do not alter the planar slip mode in the process of tensile deformation at room temperature. Therefore, the size and distribution of NbC particles are not the only reason that account for the planar slip.

Reviewer 3 Report

The authors have addressed most of the comments raised. However, I still have my concerns: 

  • As mentioned in Table 1 - Error bars are not introduced
  • Fig. 1 - X and Y-axis should be introduced with units
  • Fig. 3(b) XRD plot - Y-axis legend and units are missing. X-axis legend unit is missing. Moreover, fcc phase will have 4-5 peaks in the range considered. Why are they not visible?
  • Fig. 3(a) - typo in the scale bar should be rectified.
  • There exists still several typos that need to be rectified. For numbers with units, space should be introduced between them. For instance: 4000MPa should be written as 4000 MPa.
  • Fig. 4 - X-axis legend unit is missing.
  • Fig. 5(b) - inset - SAED pattern should be indexed carefully.
  • Scale bars in Figs. 7 and 9 are not clearly visible.

Author Response

Response to Reviewer 3 Comments

Point 1: As mentioned in Table 1 - Error bars are not introduced

Response 1: Error bars have been introduced in Table 1.

Point 2: Fig. 1 - X and Y-axis should be introduced with units

Response 2: the units have been introduced in Fig. 1.

Point 3: Fig. 3(b) XRD plot - Y-axis legend and units are missing. X-axis legend unit is missing. Moreover, fcc phase will have 4-5 peaks in the range considered. Why are they not visible?

Response 3: After another experiment, the latest experiment results have been corrected.

Point 4: Fig. 3(a) - typo in the scale bar should be rectified.

Response 4: typo in the scale bar in Fig. 3(a) has been rectified.

Point 5: There exists still several typos that need to be rectified. For numbers with units, space should be introduced between them. For instance: 4000MPa should be written as 4000 MPa.

Response 5: Have amended as required.

Point 6: Fig. 4 - X-axis legend unit is missing.

Response 6: the unit in X-axis in Fig. 4 has been added.

Point 7: Fig. 5(b) - inset - SAED pattern should be indexed carefully.

Response 7: Have indexed the inset - SAED pattern in Fig. 5(b).

Point 8: Scale bars in Figs. 7 and 9 are not clearly visible.

Response 8: Scale bars in Figs. 7 and 9 has been processed clearly.

Reviewer 4 Report

The paper has been improved. Some corrections before possible publication:

  • Please, include the tensile standard in the reference list
  • Briefly explain how elongation is defined in the standard
  • Provide further English review

Author Response

Response to Reviewer 4 Comments

Point 1: Please, include the tensile standard in the reference list

Response 1: The tensile standard has been included in the reference list.

Point 2: Briefly explain how elongation is defined in the standard

Response 2: The elongation has been defined as “where, the elongation is the percentage ratio of the total deformation of gauge length after tensile fracture to the original gauge length.”

Point 3: Provide further English review

Response 3: The English language was further review

Round 3

Reviewer 1 Report

The authors studied the mechanical properties of Fe-Mn-Al-Nb-C steel. The results can be useful for the researchers in this field. The present version has been somewhat improved from the previous one on the relation between NbC particles and planar slip.

I found a kind of typo.

p. 11, line 26
"was occurred" should be "occurred."

Please ask proof reading on English expression before submitting the final version.

Author Response

Point 1: p. 11, line 26 "was occurred" should be "occurred."
Response 1: Have changed into occurred as required

Point 2:Please ask proof reading on English expression before submitting the final version.

Response 2: The English expression has been checked and modified

Reviewer 3 Report

The authors have satisfactorily addressed my comments but still, typos exist.

Author Response

Point 1: The authors have satisfactorily addressed my comments but still, typos exist.

Response 1: The typos have been corrected